

# Cross-sectional association between 24-hour movement guidelines and depressive symptoms in Chinese university students

Yanqing Zhang[1], Xinli Chi[2], Liuyue Huang[3], Xingyi Yang[4] and Sitong Chen[5]

[1] Department of Physical Education, Xi'an University of Science and Technology, Xi'an, China
[2] School of Psychology, Shenzhen University, Shenzhen, China
[3] Department of Psychology, Faculty of Social Science, University of Macau, Macau, China
[4] School of Physical Education, Shanghai University of Sport, Shanghai, China
[5] Centre for Mental Health, Shenzhen University, Shenzhen University, Shenzhen, China

Corresponding author
Sitong Chen,
sitong.chen@live.vu.edu.au

## ABSTRACT

**Background**. The world's first 24-h movement guidelines for adults were released on 15 October 2020 in Canada, though evidence of their associations with health indicators in young adults is sparse. This study aimed to report the prevalence of meeting the 24-h movement guidelines and associations with depressive symptoms in a sample of Chinese university students.

**Methods**. Cross-sectional data from 1,793 Chinese university students (mean age = 20.7 years, 63.6% female) were used. Sociodemographic information, movement behaviors (physical activity, sedentary behavior, and sleep duration), and depressive symptoms were collected using a self-reported questionnaire.

**Results**. The prevalence of meeting the 24-h movement guidelines was 27.8% in Chinese university students. Logistic regression results show that compared to those who met the 24-h movement guidelines, odds ratio (OR) for depressive symptoms in those who met fewer recommendations contained in the 24-h movement guidelines were significantly higher (OR for none = 3.4, 95% CI [2.1–5.5], $p < 0.001$; OR for one = 2.7, 95% CI [2.0–3.8], $p < 0.001$; OR for two = 1.5, 95% CI [1.1–2.1], $p = 0.013$).

**Conclusion**. The prevalence of meeting the 24-h movement guidelines in Chinese university students was relatively low and should be enhanced through multiple strategies. Meeting the 24-h movement guidelines was associated with lower risk for depression in Chinese young adults. It is suggested that moving more, sitting less and sleeping well in this population may reduce the occurrence of depression.

## INTRODUCTION

Depression is one of the most prevalent mental health issues with persistent feeling of sadness, tiredness, and anhedonia (*i.e.,* lack of interest or happiness) (*World Health Organization, 2020*). Depressive symptoms were reported to be associated with substance abuse, eating disorders, dropout, and self-injury among university students (*De Aquino*

*Nunes et al., 2024*). Moreover, individuals who experienced deterioration in this symptom were associated with greater likelihood of suicide (*Eberhard & Weiller, 2016*), and exerts a huge health and economic burden on society (*Greenberg et al., 2003*; *Whiteford et al., 2013*). During the transition from adolescence to adulthood, university students face multidimensional challenges and pressures (*Greenberg et al., 2003*; *Whiteford et al., 2013*), making them more susceptible to psychological problems, including depressive symptoms. Data demonstrates approximately one-third of university students facing mild to severe depressive symptoms (*Garlow et al., 2008*; *Lei et al., 2016*; *El Ansari et al., 2011*). A meta-analysis study in 2016 showed that 23.8% of Chinese university students suffered from depression (*Lei et al., 2016*). This increase has been attributed to various pandemic-related stressors, including social distancing measures, disruption of routine activities, and concerns about personal health and the economy. Collectively, these data suggest that depressive symptoms are a serious public health problem in university students, which should be addressed by effective measures. Therefore, identifying protective factors against depressive symptoms in this age-group population is urgently needed.

Numerous studies have indicated that movement behaviors, including physical activity (PA), sedentary behavior (SB), and sleep are independent correlates of depressive symptoms in university students. For example, previous studies indicate that regular PA was significantly associated with reduced risk for depressive symptoms in university students (*El Ansari et al., 2011*; *Feng et al., 2014*; *Vickers et al., 2004*; *Harbour et al., 2008*). Excessive SB has been recognized as a risk factor of developing depressive symptom among university students (*Xu et al., 2020*; *Lee & Kim, 2019*). Further, insufficient sleep is also linked to greater odds of being depressed (*Feng et al., 2014*; *Carskadon et al., 2012*). As these behaviors are independent factors of depressive symptoms in university students, it is likely that increasing PA, reducing SB and encouraging appropriate sleep concurrently would be beneficial for reducing depressive symptoms. However, it still remains elusive on the combined effects of sufficient PA, limited screen time (ST) and appropriate sleep duration on depressive symptoms.

Since daily time in PA, SB, and sleep are codependent, they should be considered concurrently (*Dumuid et al., 2020*; *Pedisic, 2014*; *Pedisic, Dumuid & Olds, 2017*), making such analyses conducive to informing efficient mental health interventions (*Sampasa-Kanyinga et al., 2020*). Consequently, in October 2020, the Canadian 24-h movement guidelines for adults were released (*Ross et al., 2020*) as the world's first 24-h movement guidelines for adults. The guidelines recommend specific time durations for PA, SB, and sleep in adults (*Ross et al., 2020*), which offers a new research paradigm for better understanding 24-h movement behaviors and health. Specifically, the guidelines recommend that adults should accumulate at least 150 min of moderate to vigorous PA, limit SB no more than 8 h per day, and maintain 7 to 9 h of good-quality sleep (*Ross et al., 2020*). However, owing to their novelty, more studies using the guidelines are needed to examine their relationship with different health outcomes, such as depression, and help refine the guidelines in the future.

Although the guidelines are built on rigorous methodology and reliable evidence (*Ross et al., 2020*; *Chaput et al., 2020*; *Kho et al., 2020*; *Saunders et al., 2020*), little is known about

the prevalence of meeting the 24-h movement guidelines for adults and the association with health outcomes. For the 24-h movement guidelines for children and youth, a large number of studies have indicated that meeting more recommendations contained within the guidelines is associated with better mental health in children and adolescents (*Janssen, Roberts & Thompson, 2017*; *Carson et al., 2017*; *Lee et al., 2018*). However, research is needed to examine whether adults who meet the 24-h movement guidelines are also more likely to demonstrate better mental health. Further, a recent review indicated that there was less evidence examining 24-h movement guidelines and mental health indicators in different populations (*Rollo, Antsygina & Tremblay, 2020*). Examining the associations between the 24-h movement guidelines for adults and health is beneficial to the guidelines' further development and adoptions in a wider range of populations.

Thus, to better understand the relationship between 24-h movement guidelines for adults and mental health in university students in China (young adults), this study aimed to explore the prevalence of meeting 24-h movement guidelines and the association with depressive symptoms.

## METHODS

### Procedure and data collection

This study was a cross-sectional study (online survey), which was based on a longitudinal study (ongoing) to assess the impacts of the coronavirus disease (COVID-19) outbreak on university students' mental health (*Chi et al., 2020*). We intended to conduct at least four consecutive surveys (starting on February 2020 and performing the surveys at three-month intervals; all of the four waves of data have been collected). In this study, data from one randomly selected (wave 3, August 2020) were used because of the data availability. A convenient sampling method was adopted in the recruitment of participants in all the waves. Participants from 10 universities in Guang-dong Province were recruited in wave 3 ($n = 1,873$). For each university, we contacted one staff (some authors' colleagues) who enabled distribution of our online survey to at least five classes (regardless of majors and grades). Participants ($n = 1,159$) completed and returned the self-report questionnaire with valid data pertaining to the study variables. To increase the sample size, we also invited an additional 714 study participants from another three universities in Anhui Province and 634 completed the questionnaire. Of all the study participants, 1,793 (general response rate = 95.7%; 1,159 from universities in Guangdong and 634 from universities in Anhui) provided answers pertaining to all the variables included in this study. Study participants provided online written consent to participate in this study prior to data collection.

The survey was conducted online (https://www.wjx.cn/jq/88518635.aspx, accessed on 20/2/2021) for both convenience and safety reasons in light of the post-COVID-19 circumstance (field surveys banned for safety considerations). Both of the surveys were conducted in August (21–31st) 2020, when COVID-19 was under control by Chinese government, and students were about to go back to school for fall semester. Over a period of 10 days, students were invited to participate in the survey *via* Tencent's QQ, WeChat, Weibo, and college-related websites (such as university association websites and bulletin

board forums). Participants who had completed all questionnaires (approximately 15 min) were given 10 RMB (Chinese currency) *via* online payment (equivalent to 1.5 USD). The entire study, including recruitment, data collection, and other research procedures, were approved by the Human Research Ethics Committee (No: 2020005) of Shenzhen University.

## Measures

### Depressive symptoms (PHQ-9)

Participants' depressive symptoms were assessed by the Patient Health Questionnaire-9 (PHQ-9) consisting of nine items assessing self-report major depressive symptoms relative to the last fortnight where participants responded from 0 = not at all to 3 = almost every day (*Kroenke, Spitzer & Williams, 2001*). Previous research found that the reliability (internal consistency reliability = 0.86; test-retest reliability = 0.86) and validity (concurrent validity = 0.29; construct validity = 0.89 (Kaiser–Meyer–Olkin measure); criterion validity = 0.92 (area under the curve using receiver operating characteristic)) of PHQ-9 (Chinese version) in university students was acceptable (*Du et al., 2017*) for use in epidemiological surveys. Previous studies showed acceptable sensitivity and specificity for major depressive symptoms with the PHQ-9 score $\geq 10$ so we used a binary variable (PHQ-9 $\geq 10$) for probable depressive symptoms incidence (*Du et al., 2017*). Binary categorizations of the PHQ-9 were used in statistical analyses.

### Movement behaviors

Movement behaviors including PA, SB, and sleep were assessed *via* two self-report questionnaires, the International Physical Activity Questionnaire-Short Form (IPAQ-SF) and Pittsburgh Sleep Quality Index (PSQI). The reliability and validity of the IPAQ-SF have been reported in Chinese populations (*Macfarlane et al., 2007*), indicating the questionnaire was acceptable to capture Chinese adults' PA (intraclass correlation coefficient = 0.79; concurrent validity = 0.28) and SB (intraclass correlation coefficient = 0.97; concurrent validity = 0.27). In brief, study participants were required to recall their PA (vigorous, moderate, and light) and SB (sitting during leisure and transportation) over the past 7 days. Data processing for variables of PA and SB was consistent with the published guidelines by the IPAQ expert group (*International Physical Activity Questionnaire Research Committee, 2005*). Study participants' sleep (hours) was assessed by the PSQI that has been validated in Chinese populations (*Guo et al., 2016*). Based on the Canadian 24-h movement guidelines for adults (*Ross et al., 2020*), study participants who reported accumulating $\geq 150$ min per week of moderate- to vigorous-intensity PA (MVPA), $\leq 8$ h per day for SB, and 7–9 h per day for sleep were regarded as meeting the 24-h movement guidelines.

### Confounding variables

Based on previously published studies (*Bennie et al., 2019*; *Bennie, Teychenne & Tittlbach, 2020*), confounding variables included sex (male or female), age (years), body mass index (BMI; calculated from self-reported height (cm) and weight (kg)), siblings (single or two or more), residence (urban or rural), family structure (living with both parents, parents divorced, or other), parents' educational level (middle school or below, high school,

college or university, master or above), number of friends (none, 1–2, 3–5, 6 or more; in our context, friend refers to a person whom one knows and with whom one has a bond of mutual affection, typically exclusive of sexual or family relations), and perceived family affluence using a scale (from 0 to 10) with higher scores indicating higher perceived family affluence (*Macfarlane et al., 2007*).

## Statistical analyses

Statistical analyses were performed using STATA 16.1 (Stata Corp, College Station, Texas). We set the effect size at 2.2 (odd ratio) according to the previous study, alpha level at 0.05, and power at 0.80, leading to a required minimum sample size of 264 participants. This ensures our study is sufficiently powered to detect meaningful effects. First, descriptive statistics were determined to report frequency (percentage) and mean (standard deviation) of categorical and continuous variables, respectively. Differences in categorical and continuous variables across different groups were examined by chi square test and student t tests, respectively. Correlation coefficients among study variables were determined using Pearson (continuous variables) or Spearman correlations (categorical). To explore the association between combinations (number of recommendations met) of 24-h movement guidelines (meeting the 24-h movement guidelines as reference group) and depressive symptoms, a logistic regression model were developed. A collinearity test was performed to check whether independent variables were affected by multi-collinearity, with variance inflation factor (VIF) <10 suggested as acceptable results (*Hair et al., 2009*). For the model, depressive symptoms was treated as a binary variable (10-points of PHQ-9 as a threshold). A binary logistic regression with maximum likelihood estimation was used to assess the association between combinations of 24-h movement guidelines and depressive symptoms. Odds ratios (OR) with 95% confidence interval (CI) were reported. Statistical significance was set as $p < 0.05$ (two sided).

## RESULTS

Descriptive characteristics of study participants are presented in Table 1. The mean age of study participants was $20.7 \pm 1.6$ years. Approximately half of study participants reported meeting the PA guidelines. The prevalence meeting PA guidelines in males was 59.6% compared to 42.5% for females ($p < 0.001$). Over 70.6% reported meeting the SB guidelines (74.9% for males and 68.2% for females, $p = 0.003$). In total, 70.3% met the sleep guidelines without sex difference (male: 70.6%, female: 70.3%, $p = 0.851$). The prevalence of meeting all the three 24-h movement guidelines was 27.8% (higher in males: 37.8% than females: 22.0%, $p < 0.001$), while most study participants met one or two recommendations contained in the guidelines. Study participants reported a mean score of $6.8 \pm 5.2$ for depressive symptoms and males reported lower scores than did females (males: 6.0 *vs.* female: 7.2, $p < 0.001$). Overall, 23.2% of study participants were categorized as depressed with fewer males than females ($p = 0.001$). More detailed statistics can be found at Table 1.

Table 2 shows the bivariate correlation coefficients among the variables included in this study. Statistically significant relationships were found between meeting 24-h movement

**Table 1  Descriptive characteristics of study sample.**

| | Total | | Male | | Female | | t or chi Square | df | p |
|---|---|---|---|---|---|---|---|---|---|
| | n | % | n | % | n | % | | | |
| **Sample** | 1,793 | 100 | 653 | 36.4 | 1,140 | 63.6 | | | / |
| **Age (years)** | 20.7 ± 1.6 | | 20.8 ± 1.6 | | 20.6 ± 1.6 | | 1.59 | 1,791 | 0.110 |
| **BMI (kg/m$^2$)** | 20.3 ± 2.9 | | 21.4 ± 3.1 | | 19.6 ± 2.5 | | 12.94 | 1,118 | 0.000 |
| **Siblings** | | | | | | | | | |
| Single | 613 | 34.2 | 277 | 42.4 | 336 | 29.5 | 30.93 | 1 | 0.000 |
| Two or more | 1,180 | 65.8 | 376 | 57.6 | 804 | 70.5 | | | |
| **Residence** | | | | | | | | | |
| Urban | 1,241 | 69.2 | 430 | 65.9 | 811 | 71.1 | 18.29 | 1 | 0.000 |
| Rural | 552 | 30.8 | 223 | 34.6 | 329 | 28.9 | | | |
| **Family structure** | | | | | | | | | |
| Living with both parents | 1,261 | 90.4 | 593 | 90.8 | 1,028 | 90.2 | | | |
| Parents divorced | 109 | 6.1 | 39 | 6.0 | 70 | 6.1 | 0.30 | 2 | 0.600 |
| Other | 63 | 3.5 | 21 | 3. | 42 | 3.7 | | | |
| **Perceived family affluence** | 5.7 ± 1.6 | | 5.5 ± 1.7 | | 5.9 ± 1.6 | | −4.93 | 1,287 | 0.000 |
| Father education level | | | | | | | | | |
| Middle school or below | 867 | 48.4 | 328 | 50.2 | 539 | 47.3 | | | |
| High school | 612 | 34.1 | 221 | 33.8 | 391 | 34.3 | 6.17 | 3 | 0.291 |
| College or university | 250 | 13.9 | 76 | 11.6 | 174 | 15.3 | | | |
| Master or above | 64 | 3.6 | 28 | 4.3 | 36 | 3.2 | | | |
| **Mother education level** | | | | | | | | | |
| Middle school or below | 1,055 | 58.8 | 403 | 61.7 | 652 | 57.2 | | | |
| High school | 544 | 30.3 | 182 | 27.9 | 362 | 31.8 | 6.07 | 3 | 0.245 |
| College or university | 152 | 8.5 | 49 | 7.5 | 103 | 9.0 | | | |
| Master or above | 42 | 2.3 | 19 | 2.9 | 23 | 2.0 | | | |
| **Number of friends** | | | | | | | | | |
| None | 29 | 1.6 | 10 | 1.5 | 19 | 1.7 | | | |
| 1–2 | 590 | 32.9 | 209 | 32.0 | 381 | 33.4 | 8.68 | 3 | 0.052 |
| 3–5 | 936 | 52.2 | 327 | 50.1 | 609 | 53.4 | | | |
| 6 or more | 238 | 13.3 | 107 | 16.4 | 131 | 11.5 | | | |
| **MVPA (min/week)** | 127.0 ± 121.0 | | 152.8 ± 132.2 | | 112.2 ± 110.0 | | 9.52 | 1,050 | 0.000 |
| **Sedentary time (hours/day)** | 6.4 ± 3.7 | | 6.0 ± 3.7 | | 5.0 ± 3.7 | | −3.51 | 1,791 | 0.000 |
| **Sleep duration (hours/day)** | 7.4 ± 1.5 | | 7.3 ± 1.6 | | 7.5 ± 1.5 | | −2.52 | 1,791 | 0.012 |
| **Meeting the physical activity guidelines** | | | | | | | | | |
| No | 921 | 51.4 | 266 | 40.7 | 655 | 57.5 | 46.47 | 1 | 0.000 |
| Yes | 872 | 48.6 | 387 | 59.3 | 485 | 42.5 | | | |
| **Meeting the sedentary behavior guidelines** | | | | | | | | | |
| No | 527 | 29.4 | 164 | 25.1 | 363 | 31.8 | 9.05 | 1 | 0.003 |
| Yes | 1,266 | 70.6 | 489 | 74.9 | 777 | 68.2 | | | |

**Table 1** (*continued*)

| | Total | | Male | | Female | | t or chi Square | df | p |
|---|---|---|---|---|---|---|---|---|---|
| | n | % | n | % | n | % | | | |
| **Meeting the sleep guidelines** | | | | | | | | | |
| No | 532 | 29.7 | 192 | 29.4 | 340 | 29.8 | | | |
| Yes | 1,261 | 70.3 | 461 | 70.6 | 800 | 70.2 | 0.03 | 1 | 0.851 |
| Positive | 416 | 23.2 | 123 | 18.8 | 293 | 25.7 | | | |
| **Combinations of 24-h movement behavior guidelines** | | | | | | | | | |
| None | 109 | 6.01 | 29 | 4.4 | 80 | 7.0 | | | |
| One | 467 | 26.1 | 158 | 24.2 | 309 | 27.1 | | | |
| Two | 719 | 40.1 | 219 | 33.5 | 500 | 43.9 | 30.93 | 1 | 0.000 |
| All | 498 | 27.8 | 247 | 37.8 | 251 | 22.0 | | | |
| **Depressive symptoms** | 6.8 ± 5.2 | | 6.0 ± 5.1 | | 7.2 ± 5.2 | | −4.79 | 1,791 | 0.000 |
| **Depressive symptoms (binary)** | | | | | | | | | |
| Negative | 1,377 | 76.8 | 530 | 81.2 | 847 | 74.3 | 18.29 | 1 | 0.000 |
| Positive | 416 | 23.2 | 123 | 18.8 | 293 | 25.7 | | | |

**Notes.**

sd, Standard deviation; df, Degree of freedom; MVPA, Moderate to vigorous physical activity.

guidelines and depressive symptoms (as a binary variable; $r = -0.2$, $p < 0.001$; see Table 2).

Generalized linear models were performed to determine the association between meeting 24-h movement guidelines and depressive symptoms (as binary variable) after adjusting for the other variables, as shown in Table 3. All VIFs results were below 1.5, indicating the absence of multicollinearity. Compared with meeting the 24-h movement guidelines (meeting all the recommendations), study respondents who met fewer recommendations were more likely to report depressive symptoms. Specifically, those categorized as meeting none of the recommendations, one recommendation, or two recommendations had a higher OR for depressive symptoms in a dose–response gradient (OR for none = 3.4, 95% CI [2.1–5.5], $p < 0.001$; OR for one = 2.7, 95% CI [2.0–3.8], $p < 0.001$; OR for two = 1.5, 95% CI [1.1–2.1], $p = 0.013$). Sex-specific results are also presented in Table 3. In detail, meeting more guidelines was negatively associated with depressive symptoms in males (OR for none = 3.5, 95% CI [1.4–9.1], $p = 0.012$; OR for one = 2.6, 95% CI [1.4–4.6], $p = 0.003$; OR for two = 2.2, 95% CI [1.3–3.9], $p = 0.008$). Such association was partially observed in females but meeting two was not associated with depressive symptoms (OR = 1.3, 95% CI [0.8–1.9], $p = 0.280$).

# DISCUSSION

This study reported the prevalence of meeting the newly released Canadian 24-h movement guidelines for adults and assessed the associations between the different combinations of components of the Canadian 24-h movement guidelines for adults and depressive symptoms (assessed by PHQ-9) in a sample of university students in China. We found that <30% of Chinese young adults met the Canadian 24-h movement guidelines and meeting the guidelines was associated with lower risks for depressive symptoms.

**Table 2  Correlation matrix among the variables included in this study.**

|  | 1 | 2 | 3 | 4 | 5 | 6 | 7 | 8 | 9 | 10 | 11 | 12 |
|---|---|---|---|---|---|---|---|---|---|---|---|---|
| **1** | 1 | | | | | | | | | | | |
| **2** | 0.0 | 1 | | | | | | | | | | |
| **3** | 0.1*** | −0.1*** | 1 | | | | | | | | | |
| **4** | −0.3*** | 0.1*** | 0.0 | 1 | | | | | | | | |
| **5** | 0.1*** | 0.0 | 0.1* | −0.1*** | 1 | | | | | | | |
| **6** | −0.1* | 0.1*** | −0.2*** | 0.0 | 0.2*** | 1 | | | | | | |
| **7** | 0.0 | 0.0 | −0.1*** | 0.0 | −0.1*** | 0.1* | 1 | | | | | |
| **8** | 0.0 | −0.1*** | 0.2*** | 0.0 | −0.3*** | −0.3*** | 0.0 | 1 | | | | |
| **9** | 0.0 | −0.2*** | 0.2*** | 0.0 | −0.3*** | −0.5*** | 0.0 | 0.6*** | 1 | | | |
| **10** | −0.1 | 0.0 | 0.1*** | 0.0 | −0.0 | 0.0 | 0.0 | 0.0 | 0.1** | 1 | | |
| **11** | −0.1*** | 0.0 | 0.1*** | 0.1*** | 0.0 | 0.0 | 0.0 | 0.0 | 0.0 | 0.1** | 1 | |
| **12** | 0.1** | 0.0 | −0.1*** | 0.0 | 0.1* | 0.0 | 0.0 | 0.0 | 0.0 | −0.2*** | −0.2*** | 1 |

Notes.
*$p < 0.05$.
**$p < 0.01$.
***$p < 0.001$.
[1]Sex.
[2]Age.
[3]Perceived family affluence.
[4]BMI; body mass index.
[5]Siblings.
[6]Residence.
[7]Family structure.
[8]Father education level.
[9]Mother education level.
[10]Number of friends.
[11]Combinations of the Canadian 24-h Movement Guidelines for Adults.
[12]Depressive symptoms binary.

**Table 3  The association between meeting 24-h guidelines and depressive symptoms screened positive by generalized liner models.**

|  | Overall | | | Male | | | Female | | |
|---|---|---|---|---|---|---|---|---|---|
|  | OR | 95% CI | | OR | 95% CI | | OR | 95% CI | |
| **Meeting none** | 3.4 | 2.1 | 5.5 | 3.6 | 1.4 | 9.1 | 3.4 | 1.9 | 6.0 |
| **Meeting one** | 2.7 | 2.0 | 3.8 | 2.6 | 1.4 | 4.6 | 2.7 | 1.8 | 4.1 |
| **Meeting two** | 1.5 | 1.1 | 2.1 | 2.2 | 1.3 | 3.9 | 1.3 | 0.8 | 1.9 |

Notes.
Models controlled for sex, age, siblings, residence, family structure, father education level, mother education level, perceived family affluence, number of friends, body mass index except for age-specific models. Notes: Two = meeting two recommendations of the 24-hour movement guidelines, One = meeting one recommendation of the 24-hour movement behaviors, None = meeting none of the 24-hour movement behaviors. 24 h movement behaviors, None = meeting none of the 24-hour movement behaviors.

To our knowledge, this study is one of the first to investigate the prevalence of meeting the Canadian 24-h movement guidelines for adults in any jurisdiction, and may help advance our understanding of the relationships between movement behaviors and health in adults. The current study found that 27.8% of Chinese young adults met the 24-h movement guidelines. Owing to limited studies in this field, it is not yet possible to compare our prevalence results with other studies. However, when comparing the prevalence with

that in younger populations (children or adolescents), our study showed that Chinese young adults (at least university students) were more likely to exhibit desirable movement behaviors (a higher prevalence of meeting the 24-h movement guidelines). Specifically, for Chinese children and adolescents, a nationally representative study found that only about 5% of them met the age-appropriate 24-h movement guidelines (*Chen et al., 2021*), which is lower that our result. Another study based on American children and adolescents also showed that the prevalence meeting the 24-h movement guidelines was about 5% (*Knell et al., 2019*). Some studies reported lower prevalence of meeting the 24-h movement guidelines in children and adolescents (*Lee et al., 2019*; *Manyanga et al., 2019*; *Shi et al., 2020*). Thus, while more Chinese young adults report desirable movement behaviors compared to younger populations, the proportion of meeting these minimal guidelines remains unacceptably low. It should also be noted that the large difference between young adults and children and adolescents might result from survey methodology and survey timing. Concerning the measurements, in our study, IPAQ and PSQI were used to assess movement behaviors while other studies based on children and adolescents used the Health Behavior in School-aged Children (HBSC) questionnaire (*Chen et al., 2021*; *Chen & Yan, 2020*) or accelerometers to collect data of MVPA, SB, and sleep (*Shi et al., 2020*; *Roman-Viñas et al., 2016*). The difference in the measurements could be explain by the difference in prevalence between young adults, and children and adolescents. Another explanation involves the survey timing. Data collection for the current study was conducted in August, summertime for university students, when they may be more likely to engage in more PA and sleep longer because of less academic pressure and loads. This would in turn lead to limited time spent in SB, resulting in a higher prevalence meeting the 24 movement guidelines in university students compared with children and adolescents who participated in the survey during academic semester (*Chen et al., 2021*). Of note, it should be mentioned that owing to COVID-19 preventive policies, some study participants' PA, like outdoor time or leisure activities may be affected. In this regard, our study might be biased to reflect study participant' regular movement behavior. Future studies are encouraged to monitor 24-h movement behaviors throughout a longer period and across the year to examine the stability of 24-h movement behaviors.

When looking at the prevalence meeting the PA, SB, and sleep recommendations individually, the prevalence meeting the PA recommendation was the lowest in the present study and consequently had the greatest impact on the prevalence of meeting the 24-h movement guidelines. This finding is similar to some studies on children and adolescents (*Chen et al., 2020*; *Shi et al., 2020*; *Roman-Viñas et al., 2016*). Young adults, especially university students, often achieve insufficient MVPA (*Zurita-Ortega et al., 2018*; *Staiano et al., 2018*; *Unick et al., 2017*), which has been well documented in epidemiological surveys. Based on the current study, the promotion of habitual MVPA for Chinese young adults' will be likely help achieve desired 24-h movement behaviors. More research is also required to understand young adults' time-use pattern and determine the main determinants of the low prevalence of meeting the 24-h movement guidelines (*Chen et al., 2021*; *Chaput et al., 2014*). More time-use epidemiological studies in young adults are needed (*Dumuid et al., 2020*; *Pedisic, Dumuid & Olds, 2017*).

Many previously published studies have suggested that meeting the 24-h movement guidelines is associated with better health outcomes, including mental health indicators in young populations (*Janssen, Roberts & Thompson, 2017*; *Lee et al., 2018*; *Carson et al., 2019*). Our study, as one of the very first to assess the 24-h movement guidelines and mental health in young adults, adds evidence on the association between 24-h movement guidelines and depressive symptoms in young adults, which could inform strategies to promote mental health. The present study found that meeting the 24-h movement guidelines presented the lowest risks for depressive symptoms in university students compared to meeting fewer recommendations of 24-h movement guidelines. To the best of our knowledge, there is no comparable research evidence on adults to support or negate our findings. However, evidence from research based on younger populations does provide support. For example, studies by *Carson et al. (2017)* and *Janssen, Roberts & Thompson (2017)* indicated that meeting more recommendations contained within 24-h movement guidelines were associated with better mental health in children and adolescents. In US adolescents, meeting the 24-h movement guidelines was significantly associated with reduced depressive symptoms (*Zhu, Haegele & Healy, 2019*). Similar research findings were also found in a study consisting of South Korean adolescents (*Lee et al., 2018*). Owing to the limited evidence based on studies of adults, in part because the 24-h guidelines for adults are very new, more studies are encouraged to explore the association between 24-h movement guidelines and depressive symptoms, and other health outcomes, in adults of varying ages. In addition to this, a large body of evidence has suggested that sufficient MVPA (*El Ansari et al., 2011*; *Feng et al., 2014*; *Vickers et al., 2004*; *Harbour et al., 2008*), limited SB (*Xu et al., 2020*; *Lee & Kim, 2019*) and appropriate sleep (*Feng et al., 2014*; *Carskadon et al., 2012*) were separate correlates of depressive symptoms. Hence, it is reasonable to expect that their combinations would lead to cumulative effects on reducing depressive symptoms. Evidence from studies using compositional data analysis provide further support to our findings (*Dumuid et al., 2020*; *Zhu, Haegele & Healy, 2019*). Studies based on adults have indicated that replacing SB with PA or sleep would be associated with positive mental health outcomes (*Larisch et al., 2020*; *Kitano et al., 2020*). This suggests that people with optimal time reallocation for compositions of movement behaviors could gain health benefits. Comparatively, meeting the 24-h movement guidelines could be viewed as a desirable status to achieve to be associated with better mental health outcomes. However, the underlying mechanism(s) linking 24-h movement behaviors to depressive symptoms should be clarified in the future.

An interesting finding is that the association between 24-h movement guidelines and depressive symptoms varied by sex, as in females, meeting any two of the 24-h movement guidelines does not show significantly higher lower odds for depressive symptoms compared to meeting all three guidelines. However, this finding was not found in males. This difference could not be explained in our study, as the mechanism interpreting the sex difference remains unclear. It is, therefore, to explore the sex difference in the association in the future. Of note, this research finding is beneficial to design sex-specific interventions aiming at reducing depression symptoms while through movement behavior changes.

Based on research findings of the current study, some practical implications can be proposed. To prevent mental disorders (*e.g.*, depressive symptoms) of university students, encouragement for more engagement in PA, restrictions of SB, and appropriate sleep would be a beneficial approach. It is needed and recommended to consider the integration of PA, SB, and sleep when designing interventions aiming to reduce or prevent depressive symptoms in university students —a "whole day matters" approach. For university students, schools and families are the two settings in which they spend most of their time, it is therefore recommended to design effective strategies linked with schools and families. However, it should be encouraged to design sex-specific strategies for young adults with different sexes.

Although this study is one of the first to assess the prevalence of meeting the 24-h movement guidelines in young adults and the association between 24-h movement guidelines and depressive symptoms, some inherent study limitations should be mentioned. First, owing to the cross-sectional nature of the study, we cannot infer causality for the association between 24-h movement guidelines and depressive symptoms. Second, self-reported questionnaires were used to assess 24-h movement behaviors in our study, which could result in measurement errors. Third, our study recruited samples using a convenient sampling procedure instead of random sampling, which reduces the representativeness; so, the research findings may not be generalizable into other similar populations. Fourth, the Canadian 24-h guidelines also recommend that within the SB period, recreational screen time should not exceed three hours. Future research should also incorporate this recommendation. Fifth, there are many factors related with depressive symptoms in university students. However, in our study, only a limited number of confounding variables were included. Sixth, for the measures of family structure, we failed to collect more details and information on it, which may result in some bias in estimating the association between independents and dependents. Seventh, our study's region targeted a few provinces, which limits the generalizability of our findings to all university students to the whole China. Finally, owing to study survey period was during the COVID-19 pandemic, some factors would have influences on 24-h movement behaviors and depressive symptoms, which may not be considered in our study. Such omissions could result in bias to our study results. It should be, therefore, extremely cautious to interpret our results.

## CONCLUSIONS

This study reports the prevalence of meeting the 24-h movement guidelines for adults and assessed the association with depressive symptoms in university students (young adults). We observed that in this sample less than 30% of Chinese university students met the 24-h movement guidelines and that meeting more recommendations within the guidelines had lower odds for depressive symptoms. The current study provides preliminary evidence to support the concurrence of sufficient PA, limited SB, and appropriate sleep associated with better mental health. For university students' mental health, it is recommended to make them move more, sit less and sleep well. Further research is recommended to confirm these findings and to explore associations between adult 24-h movement behaviors and other mental health outcomes.

## ACKNOWLEDGEMENTS

We would like to thank the participants in this study.

### Funding

This work was supported by the National Social Science Foundation of China (grant number 16CSH049), and Guangdong Basic and Applied Basic Foundation; Yang Liu was supported by Shanghai Sport Science Project (21Q007) from Shanghai Administration of Sports. The funders had no role in study design, data collection and analysis, decision to publish, or preparation of the manuscript.

### Grant Disclosures

The following grant information was disclosed by the authors:
The National Social Science Foundation of China: 16CSH049.
Guangdong Basic and Applied Basic Foundation.
Shanghai Sport Science Project from Shanghai Administration of Sports: 21Q007.

### Competing Interests

The authors declare there are no competing interests.

### Author Contributions

- Yanqing Zhang conceived and designed the experiments, performed the experiments, prepared figures and/or tables, authored or reviewed drafts of the article, and approved the final draft.
- Xinli Chi conceived and designed the experiments, performed the experiments, analyzed the data, prepared figures and/or tables, authored or reviewed drafts of the article, and approved the final draft.
- Liuyue Huang conceived and designed the experiments, performed the experiments, prepared figures and/or tables, authored or reviewed drafts of the article, and approved the final draft.
- Xingyi Yang analyzed the data, prepared figures and/or tables, authored or reviewed drafts of the article, and approved the final draft.
- Sitong Chen conceived and designed the experiments, performed the experiments, analyzed the data, prepared figures and/or tables, authored or reviewed drafts of the article, and approved the final draft.

### Human Ethics

The following information was supplied relating to ethical approvals (i.e., approving body and any reference numbers):

Recruitment and data collection procedures were approved by the Human Research Ethics Committee (No:2020005) of Shenzhen University.

## Data Availability

The raw measurements are available in the Supplementary File.

## Supplemental Information

Supplemental information for this article can be found online at http://dx.doi.org/10.7717/peerj.17217#supplemental-information.

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
