# Peer review of "Cross-sectional association between 24-hour movement guidelines and depressive symptoms in Chinese university students"

_PeerJ, doi:10.7717/peerj.17217_

## Round 0.1 · original submission · Major Revisions

Please, address point-to-point all reviewers' issues, in particular Reviewer 1 and 2's.

**Language Note:** PeerJ staff have identified that the English language needs to be improved. When you prepare your next revision, please either (i) have a colleague who is proficient in English and familiar with the subject matter review your manuscript, or (ii) contact a professional editing service to review your manuscript. PeerJ can provide language editing services - you can contact us at copyediting@peerj.com for pricing (be sure to provide your manuscript number and title). – PeerJ Staff

Reviewer 1 ·

Basic reporting

Line 40-42:
“Depressive symptoms were reported to associate with substance abuse, eating disorders, dropout, and self-injury among university students [2].”

Comments: “reported to associate” should be replaced as “reported to be associated”.

Line 40-42:
“Depressive symptoms were reported to associate with substance abuse, eating disorders, dropout, and self-injury among university students [2].”

Comments: This reference here is outdated and is a narrative review. Please provide a recent reference, preferably epidemiological evidence or a meta-analysis for each health outcome and depressive symptom.

Line 47-48:
“A meta-analysis study in 2016 showed that 23.8% of Chinese university students suffering from depression [7].”

Comments: This reference is crucial. However, current research used data during the COVID-19 period. Please provide some statements on the prevalence of depression during the COVID-19 period.

Line 54-55:
“For example, previous studies indicate that regular PA was significantly associated with reduced risk for depressive symptoms [8-11].”

Comments: What is the population of this statement?

Line 60-61:
“However, it still remains elusive on the combined effects of sufficient PA, limited ST and appropriate sleep duration on depressive symptoms.”

Comments: What does the “ST” mean? Is that the abbreviation of sedentary behavior? Please double-check the manuscript and ensure that abbreviations are consistent throughout the manuscript.

Experimental design

Line 93:
“In this study, data from wave 3 were used.”

Comments: Please provide the year and month of wave 3.

Line 113-115:
“Recruitment and data collection procedures were approved by the Human Research Ethics Committee (No:2020005) of Shenzhen University.”

Comments: This sentence needs to be modified. The whole study rather than only recruitment and data collection should be approved by the local ethical committee.

Validity of the findings

Line 173-174:
“The prevalence meeting PA guidelines in males was 59.6% compared to 42.5% for females (p = 0.000).”

Comments: “p=0.000” should be reported as “p<0.001”, please check this through the whole manuscript.

Line 175-177:
“The prevalence of meeting the 24-h movement guidelines was 27.8% (higher in males: 37.8% than females: 22.0%, p = 0.000), while most study participants met one or two recommendations contained in the guidelines.”

Comments: It should be “The prevalence of meeting all the three 24-h movement guidelines was 27.8%”.

Additional comments

None.

Reviewer 2 ·

Basic reporting

Line 48, where the citation is from 2016, suggest adding some new references.

Experimental design

1.The novelty of the theme is high, but it also has flaws. The title needs to be adjusted for novelty.
2.Is it convincing why the authors chose college students in Anhui and Guangdong to represent all college students in China? The article needs to explain the region where the sample size is located. Is the title able to cover all areas? It is suggested that the parts of the country be refined.

Validity of the findings

1.Do the questionnaires distributed after the epidemic have an impact on the psychological situation and 24-hour behavior of university students?
2. It is recommended to rewrite the content of this section as a 24-hour behavior habit for adults and minimize the guideline standards for children and adolescents in the article. It is suggested to analyze the reasons from the living situation of Chinese college students. In addition, it is suggested to add a post-pandemic perspective to the discussion to explain the degree of depression and 24-hour behavioral habits of college students.

Additional comments

The abstract lacks specifics on the results, for example, 24-hour behavioural satisfaction of Chinese, and specific odds ratios (ORs).

Reviewer 3 ·

Basic reporting

Professional and scientific language use, with evident of well-structured paper format and use of references meeting PeerJ's standard. Tables are displayed in appropriate format whilst no figure is contained in the manuscript (not needed from a reviewer's perspective).

Experimental design

The authors did well in setting a good scene for this research; that is, they clearly articulated background issues and aims, reviewed in-depth literature, and provided a well-justified rationale for the study. The methodological approach, despite some limitations (i.e., cross-sectional nature, self-report measures), was compensated by other noticeable features (e.g., large sample, novelty, robust statistical control, etc). However, I do have several minor concern that would like to ask the authors for more clarifications:

1. Whilst it was mentioned the study participants/data were from the 3rd wave of an ongoing longitudinal project, it is unclear to readers why the specific wave of data was chosen.
2. Also, given the longitudinal nature of the original project and the acknowledged limitation of cross-sectional design, does the longitudinal project not having another time point assessing movement patterns or depressive symptoms? The authors would do well to provide more justifications to the approach they took.
3. The sample size of the study was good; however, it lacked information on what effect size of the sample could be considered 'sufficient' in detecting power. The authors, therefore, would do well to provide a power analysis/sample estimation to suggest what OR the sample was good to detect (i.e., at least .80 power).
4. Please provide a reference to back up the decision of accepting VIF <10 (some people would argue VIF not excessing 5), though not a major issues as the authors reported all VIFs < 1.5.
5. When assessing the extent to which participants meet the 24-h movement pattern, the authors aggregate PA, SB, & Sleep pattern to 4 levels of the 24-movement pattern (i.e., meeting none, meeting 1 guideline, meeting 2 guidelines, meeting all guidelines). Such an approach was conventional but failed to tackle potential divergence between different combinations of a pair of two guidelines (e.g., meeting PA/SB vs meeting PA/Sleep vs meeting SB/Sleep). The authors would do well to justify why they couldn't examine the different combinations of meeting 2 guidelines, as one would argue that adopting such an approach could provide information about what to prioritise among the three guidelines. The authors should at least considering discussing this as a limitation.

Validity of the findings

All statistical analyses were performed appropriately and the findings were reported and interpreted nicely. One minor point appeared on p.18, lines 278-280 - shouldn't meet any two guidelines....associated with higher (not lower) odds ...compared with meeting all three guidelines?

Additional comments

Occasionally, in the manuscript, the authors interpreted the findings and referred it to "young adults". However, I believe the authors should keep the term as in their title and refer to "university students or college students" as young adults could be graduates who are involved in full-time or part-time jobs, which adds a different level of confounders to depressive symptoms. Being university students, not working population, may also explain the relatively high proportion of meeting all three guidelines in the current study sample when compared to similar studies in other countries.

---

## Round 0.2 · accepted · Accept

Authors addressed all reviewers' issues well enough.

Reviewer 1 ·

Basic reporting

The authors addressed my question very well and I recommend that this article should be accepted for publication.

Experimental design

no comment

Validity of the findings

no comment

Reviewer 3 ·

Basic reporting

The authors have addressed all the potential concerns and suggested changes relating to the basic reporting aspect of the manuscript.

Experimental design

The authors have addressed all the potential concerns and suggested changes relating to the experimental design aspect of the manuscript.

Validity of the findings

The authors have addressed all the potential concerns and suggested changes relating to the validity of the findings aspect of the manuscript.

Additional comments

The authors have done an excellent job in considering and responding to all of the reviewer feedback. I feel that they have done a thorough job with the revisions, and any suggestions I have at this point would reflect more of a personal writing style than substantive content. Therefore, I respect the authors' choices and am pleased with the revised manuscript. Good work!